# The Impact of COVID-19 and the Pandemic on Tinnitus: A Systematic Review

**DOI:** 10.3390/jcm10132763

**Published:** 2021-06-23

**Authors:** Eldre Beukes, Alyssa Jade Ulep, Taylor Eubank, Vinaya Manchaiah

**Affiliations:** 1Department of Speech and Hearing Sciences, Lamar University, Beaumont, TX 77710, USA; aulep@lamar.edu (A.J.U.); teubank@lamar.edu (T.E.); vinayamanchaiah@lamar.edu (V.M.); 2Vision and Hearing Sciences Research Centre, Faculty of Science and Engineering, Anglia Ruskin University, Cambridge CB1 1TP, UK; 3Department of Speech and Hearing, School of Allied Health Sciences, Manipal University, Karnataka 576104, India

**Keywords:** COVID-19, coronavirus, tinnitus, hearing loss, vertigo, systematic review, auditory symptoms, pandemic

## Abstract

This review aimed to systematically review what has been published regarding tinnitus during the coronavirus disease 2019 (COVID-19) pandemic up to March 2021 by performing both narrative and quantitative meta-analyses. Of the 181 records identified, 33 met the inclusion criteria, which generally had a fair risk of overall bias. In the included, 28 studies focused on the impact of the COVID-19 virus on tinnitus and 5 studies focused on the impact of the pandemic on tinnitus. From the studies identifying the impact of COVID-19 on tinnitus, there were 17 cross-sectional studies (*n* = 8913) and 11 case series or case report studies (*n* = 35). There were 2 cross-sectional studies (*n* = 3232) and 3 pre-post-test design studies (*n* = 326) focusing on the impact of the pandemic on tinnitus. No consistent patterns were found regarding the presentation of the tinnitus or additional factors that could have tinnitus developing in the disease impact studies. For the pandemic impact studies, the associated stress and anxiety of the pandemic were consistently suggested to contribute to tinnitus experiences. The pooled estimated prevalence of tinnitus post COVID-19 was 8% (CI: 5 to 13%). Medical professionals should be aware that tinnitus might be more problematic following the pandemic or after having COVID-19.

## 1. Introduction

On 11 March 2020, weeks after discovering a rapidly spreading Severe Acute Respiratory Syndrome Coronavirus 2 (SARS-CoV-2), the World Health Organisation (WHO) declared the coronavirus disease 2019 (COVID-19) outbreak a global pandemic [1]. A wide range of symptoms has been associated with contracting COVID-19, including respiratory failure, fever, headaches, and loss of taste and smell [2]. The severity of these symptoms ranges from being asymptomatic to having fatal consequences [3]. In addition, auditory-related conditions such as dizziness, tinnitus, and otalgia have been identified as common COVID-19 symptoms [4]. The duration of the symptoms also varies from being acute (lasting up to 4 weeks), ongoing (lasting 4–12 weeks), or lasting more than 12 weeks, referred to as “long COVID” [4]. According to the National Institute for Health and Care Excellence (NICE) [4], common symptoms of long COVID include dizziness, tinnitus, and otalgia. This is plausible since several viral infections have been identified to directly damage the inner ears; increase susceptibility to fungal or bacterial infections; or induce inflammatory responses such as measles, rubella, and cytomegalovirus [5,6].

The prevalence of audio-vestibular symptoms following the contraction of COVID-19 has been estimated by numerous systematic reviews. The first, by Almufarrij et al. [7], was published from searches in May 2020, followed by one by Saniasiaya [8] and Maharaj et al. [9] regarding searches in July 2020. A systematic review undertaken in December 2020 investigating audio-vestibular symptoms following contracting COVID-19 indicated that tinnitus had an estimated prevalence of 14.8% (CI: 6.3 to 26.1) from 12 studies [10]. Other audio-vestibular symptoms were less prevalent, such as hearing loss (7.6%; CI: 2.5–15.1) and vertigo (7.2%; CI: 0.01–26.4). A further review by Jafari et al. [11] indicated a lower prevalence range (4.5%; CI: 1.2 to 15.3) from six studies.

Due to the alarming spread of the virus through human-to-human transmission, many countries enforced regional lockdowns to reduce social interactions [12]. Although these measures reduced the spread of the virus, the restrictive measures imposed lead to a negative impact on wellbeing and increased mental health difficulties in the general population [13,14,15,16]. Certain populations were identified as being at higher risk of the pandemic negatively impacting them. This included those with tinnitus, due to the bidirectional relationship between stress and tinnitus, resulting in tinnitus being initiated or exacerbated during stressful periods [17]. As tinnitus is known to impact people differently, the effect of the pandemic on pre-existing tinnitus was unknown. Initial studies reported a range of outcomes such as some individuals finding tinnitus to be stable and others finding it worse (e.g., Beukes et al. [18]).

The systematic reviews to date have helped identify estimates of tinnitus and other auditory-vestibular dysfunctions. There is, however, not much known about the presentations of tinnitus, which will be further explored by this review. This is in aid of identifying possible risk factors, patterns in the tinnitus presentations, the tinnitus onset post-infection, and whether it resolves or changes. As no review has focused specifically on tinnitus or incorporated the effect of the pandemic on tinnitus, this review aimed to include these effects. The specific aims were to (i) investigate the effect of contracting COVID-19, (ii) determine the impact of the pandemic on tinnitus experiences, (iii) identify the progression and characteristics of the tinnitus, and (iv) comprehensively evaluate factors that could contribute to understanding the association between COVID-19 and tinnitus.

## 2. Materials and Methods

### 2.1. Protocol and Registration

This systematic review was prospectively registered with the International Prospective Register of Systematic Reviews (PROSPERO number CRD42021235661, registered on 10 February 2021) where the protocol can be found. No changes were made after registration to the protocol. The methods selected were guided by the Preferred Reporting Items for Systematic reviews and Meta-analyses (PRISMA; see Appendix A) [19]. As this was a review, registration with an institutional review board was not required. 

### 2.2. Eligibility Criteria

Eligibility was determined according to the PCECOS Criteria (Table 1). The population of interest was those experiencing tinnitus during the COVID-19 pandemic or due to COVID-19. Populations describing other audiological symptoms without any tinnitus were excluded. The primary outcome was tinnitus associated with the COVID-19 virus and the COVID-19 pandemic. Secondary outcomes were reporting any other hearing-related symptoms such as hearing loss or vestibular complaints. Any interventions or diagnostic tools managing COVID-19 or the effects of the pandemic were included. All studies (cohort, cross-sectional, case report, case-control studies, and commentaries), irrespective of the study design, were included but systematic reviews were excluded. Unpublished data, pre-prints, and secondary publications of the main published paper were excluded. All language publications were included with no date restrictions.

### 2.3. Information Sources

The following electronic research databases were used: PubMed (MEDLINE), Cumulative Index to Nursing and Allied Health Literature (CINAHL), Academic Search Complete, and EBSCOhost including Web of Science. Additional searches included hand-searching key journals and the reference lists from the included studies, citation tracking, and grey literature in Google Scholar. Unpublished data, including preprints, were excluded.

### 2.4. Search Strategy

A peer-reviewed search strategy was developed and tested through an iterative process. The keywords ‘tinnitus’ AND ‘COVID-19′ OR ‘coronavirus’ were used for searching. The search terms were used with Boolean operators and in combination with MeSH terms for each search engine to maximize outputs from the literature search. The searches were re-run until 31 March 2021, before the final analysis to ensure that no relevant articles were missed. Appendix A provides the search strategy results, including the number of records returned. Three authors (EB, AU, and TE) independently searched the databases and screened the studies to identify which met the inclusion criteria by viewing the abstracts between 15 and 20 February 2021. Periodically, until submission, searches were redone during the review process to assess for any further studies up to 31 March 2021. Included studies were cross-referenced with previous related reviews.

### 2.5. Data Management and Study Selection 

The records were exported to Rayyan [20] for independent blinded eligibility screening by three reviewers (EB, AU, and TE). Duplicate records were identified and manually removed. The title and abstract were screened, and the full text was inspected when required. For records passing the initial screen, the full texts were subsequently read to determine eligibility.

### 2.6. Data Collection Process and Data Items

For each study, relevant data suggested by the PRISMA were extracted onto a data extraction Excel spreadsheet designed by the researchers for purposes of this review. The manuscripts were divided for extraction by three authors (AU, TE, and EB) and cross-checked by each other. Initially, descriptive data were extracted regarding the reference, country, population, sample size, study design, mean age, and gender ratios. In addition, the following outcomes were extracted: reports of tinnitus, reports of tinnitus changes, and reports of other audio-vestibular difficulties (see Table 2, Table 3 and Table 4). Symptoms of dizziness, disequilibrium, and balance problems were classified as vestibular disorders.

### 2.7. Risk of Bias Assessment

Due to the different types of study designs included in this review, quality assessment for the included studies was assessed using the National Institute of Health Quality Assessment Tools [53]. Although other tools are available, using the same tool as used in similar systematic reviews (e.g., Almufarrij and Munro [10]) allowed for consistency. Specifically, the Quality Assessment Tool for Case-Control Studies, for Observational Cohort and Cross-Sectional Studies, and for Before-After (Pre-Post) Studies with No Control Group were used depending on the study design. The included studies were assessed for risk-of-bias following the 9–14 questions on each checklist (see Appendix A). Each item was judged blinded and independently by two reviewers (AU and TE). These ratings were compared and verified by a third reviewer (EB). An overall quality rating was made as good (unbiased and fully described), fair (unbiased results despite missing data), or poor (substantial details missing or questionable results).

### 2.8. Strategy for Data Synthesis

This review focused on synthesizing factors that may contribute to the presence of tinnitus by using a formal narrative synthesis as described by Campbell et al. [54] and Popay et al. [55]. The synthesis was conducted independently by three reviewers, and the combined agreed results were reported. Meta-analysis was conducted to pool the prevalence of tinnitus from the cross-sectional studies. The pooled estimates and 95% CI were computed using Comprehensive Meta-Analysis version 3 [56]. The model selected (fixed or random-effect) would depend on statistical heterogeneity. If *I*^2^ is high (larger value), indicating that effect sizes vary across the included studies, a random-effect model would be used to pool the data [57]. The results will be presented in a Forest Plot.

### 2.9. Subgroup Analysis 

Subgroup analysis of the included studies included those describing the disease impact of COVID-19 on tinnitus and those looking at the impact of the pandemic on tinnitus. Subgroup analysis was then done depending on study design, i.e., cross-sectional, pre-/post-test designs, or case studies/case controls.

## 3. Results

### 3.1. Study Selection

Database searching identified 181 retrieved records. After removing duplicates, 65 records were screened for inclusion. Of these, 33 studies met the inclusion criteria (see Figure 1). Potential studies were most often excluded due to not fulfilling the criteria of outcomes and study design or being a pre-print and not yet published. All studies included were published in 2020–2021 with data collection between January and October 2020. Most studies were specific to a single country, including regions of China, Brazil, Qatar, Germany, Turkey, the United Kingdom, Malaysia, Egypt, Turkey, Italy, Iran, India, and the United States. There were two international studies [18,33] and one regional study in Europe [41]. All the studies were in English, except one which was in Russian [32]. A translated copy was obtained to include in this review. Where numbers were not clearly stated regarding individuals with tinnitus, the study authors were contacted for clarification (e.g., Davis et al. [33]).

### 3.2. Study Characteristics

Due to the variation in the studies included in this systematic review, they were grouped initially by research question. There were 28 studies investigating the impact of COVID-19 disease on tinnitus and 5 studies investigating the impact of the COVID-19 pandemic on tinnitus. Due to the heterogeneity of the studies investigating tinnitus initiation following contracting COVID-19, the studies were further grouped into case reports and case series studies or cross-sectional studies. Among the disease impact studies, there were 11 case series/reports (Table 2) and 17 cross-sectional studies (Table 3). Among the pandemic impact studies, there were two cross-sectional, and three pre-/post-test study designs (Table 4). Findings from these studies are summarized in the next sections.

### 3.3. Risk of Bias in the Individual Studies

The quality assessment analyses of individual included studies are provided in Table 2, Table 3 and Table 4 and Appendix A. Overall, the study designs included were of low quality relative to the hierarchy of evidence in trials as no randomized controlled trials were included. The quality of the included studies was, however, fair in most cases (*n* = 25, 78%), with 4 (12.5%) being rated good, and 3 (9.5%) studies being rated poor, generally due to lacking details. The included studies thus generally provided unbiased accounts of tinnitus descriptions. The results of the individual studies are presented in the next sections.

### 3.4. Case Reports/Case Series Disease Impact Studies 

#### 3.4.1. Study Characteristics

There were 11 case reports documenting the onset or aggravation of tinnitus, sometimes reported together with other audio-vestibular symptoms (see Table 2). There were 35 cases in total with 9 case studies, 20 cases by Cui et al. [22], and 6 by Karimi-Galougahi et al. [25]. Most studies were specific to a single country, including Germany [23], the State of Qatar [21], United Kingdom [26], Ireland [28], Brazil [27], Turkey [24], Malaysia [29], Egypt [30], China [22,31], and Iran [25]. There was great variability in the ages of the patients, with the youngest being 23 years and the oldest being 67 years, with an overall mean of 42 years. Of the 14 patients with tinnitus, 6 were male (43%) and 8 were female (57%). 

#### 3.4.2. Pre-Existing Health Conditions

Most studies reported no pre-existing head trauma, ototoxic medication, or hearing disorders. Pre-existing health conditions were described in three studies, including mediated rheumatoid arthritis [27], medicated asthma [26], diabetes, hypertension, and Meniere’s disease [22]. Five studies reported no relevant comorbid diseases [24,25,28,29,30], and comorbidities were not described in three studies [21,23,31]. Hence, a range of medical backgrounds was found for these case studies. 

#### 3.4.3. Tinnitus Characteristics

In total, 14 patients (40%) reported tinnitus in the case reports included in this review. Few of the case reports provided clear descriptions of the tinnitus experienced. Where provided, great variability was found, for example, a 4 kHz and 10 dB sensation level using a tinnitus evaluation [21]; loud, white noise in both ears [23]; non-pulsatile [29]; disabling [27]; and gradually worsening [30]. There was no consistency regarding the location of the tinnitus, reported bilaterally [23], right-sided [28,29], and left-sided [21,23]. The remaining three case reports [22,24,25] reported aggravation or onset of tinnitus during COVID-19 without any descriptive information. Chirakkal et al. [21] was the only study that utilized a tinnitus evaluation comprised of frequency and intensity matching. 

#### 3.4.4. Tinnitus Initiation

The exact timings of the tinnitus initiation post-COVID-19 were furthermore lacking. Chirakkal et al. [21], Fidan [24], Lamounier et al. [27], Maharaj and Hari [29], and Sun et al. [31] reported the onset of tinnitus with the diagnosis of COVID-19. Degen et al. [23] reported tinnitus alongside deafness after the patient’s recovery following thirteen days in the intensive care unit (ICU) for COVID-19, and Koumpa et al. [26] reported tinnitus a week after transferring out of the ICU. Lang et al. [28] reported tinnitus onset after recovery from COVID-19. The remaining studies were unclear regarding the onset of tinnitus.

#### 3.4.5. Tinnitus Persistence or Recovery

Only two of the studies mentioned recovery of tinnitus. One study was two months post-recovery [29], and the other mentioned alleviation of dizziness and tinnitus following treatments with betahistine, a dihydrochloride tablet often used to treat vertigo symptoms [22]. Other studies reported tinnitus to persist post-recovery [21,27]. The remaining studies did not elaborate on tinnitus duration. Thus, a need for follow-up assessments regarding the recovery or persistence of tinnitus can aid in the understanding of the impacts of COVID-19 disease and treatment on tinnitus.

#### 3.4.6. Hearing Loss

One patient reported conductive hearing loss in the right ear [24], and eight patients reported sudden sensorineural hearing loss as a potential COVID-19-related symptom (total *n* = 12, 33%) [21,23,24,25,26,27,28,30,31]. Bilateral hearing loss was found in two patients [23,27], and 10 presented unilateral hearing loss, with 5 in the right ear [21,25,28], 4 in the left ear [23,25,26,30], and one patient presented unspecified hearing loss [31]. 

Pre-existing hearing loss was described in some studies, with only one patient presenting with hearing loss before coronavirus confirmation [27]. Lamounier et al. [27] reported audiological testing prior to the pandemic revealing isolated hearing loss at frequencies 6 and 8 kHz in the right ear only with thresholds being 45- and 30-dB HL, respectively. Audiological outcome measures to confirm hearing loss after the contraction of COVID-19 varied and included pure tone audiometry (air- and bone conduction), pure tone audiometry (bone-conduction only), acoustic immittance, speech audiometry, otoacoustic emissions, acoustically evoked potentials, and bedside testing with tuning forks. Variability in outcome measures yielded diverse reporting measures of audiological testing. Diagnostic imaging was furthermore utilized in some studies to aid in the confirmation of hearing loss. For instance, Degen et al. [23] reported magnetic resonance imaging (MRI) findings of the right and left cochlea revealing inflammation of the meninges and the right cochlea, consistent with a diagnosis of a dead right ear. Following diagnosis, the stability and management of hearing loss were unclear in most studies. Management of hearing loss was discussed in a few studies, such as via medication, corticosteroid therapy (the most common), and amplification. Where provided, three studies reporting the use of corticosteroids revealed improvement [26,27,30] and one study revealed no improvement in hearing sensitivity [34]. For example, isolated improvements in hearing following combined corticosteroid therapy (oral and intratympanic) were reported in Lamounier et al. [27] at 0.25 kHz in the right ear (from 60 dB, the threshold became 15 dB) and at 4, 6, and 8 kHz in the left ear (the thresholds became 15 dB, 5 dB, and 20 dB, respectively). Management of hearing loss using amplification, specifically cochlear implantation, was reported in only one study [23] following MRI findings indicative of inflammatory processes in the cochlea. Due to concerns regarding soft tissue formation or ossification, which could hamper surgical insertion of the electrode, urgent implantation was recommended.

#### 3.4.7. Vestibular Impairment

Vestibular difficulties associated with coronavirus were reported in only three patients (8%), all with positive results when tested for the coronavirus. Information regarding vestibular dysfunction was limited. Cui et al. [22] reported tinnitus and dizziness for a 52-year-old male with a history of diabetes and Meniere’s disease, which was alleviated with betahistine, a commonly prescribed drug for balance disorders used to alleviate vertigo symptoms. Due to the coexistence of Meniere’s disease, which manifests such symptoms with coronavirus, it is difficult to determine a connection between the virus, dizziness, and tinnitus in this case report. Treatment, stability, or recovery were not discussed. Lastly, Maharaj and Hari [29] presented a 44-year-old male admitted to the hospital after experiencing acute onset of spontaneous vertigo with nausea/vomiting and right-sided non-pulsatile tinnitus. His hearing was in the normal range and bedside vestibular testing and caloric testing revealed weakness in the semi-circular canal. Specifically, a tendency to fall towards the right side and associated horizontal torsional spontaneous nystagmus beating toward the unaffected side was reported. Management or follow-up was not discussed in the study.

#### 3.4.8. COVID-19 Testing

COVID-19 testing (positive or negative) information was included in most studies, with reverse transcription-polymerase chain reaction (RT-PCR) test being the most used. Three studies used both RT-PCR and radiographic imaging to diagnose coronavirus [24,25,31]. Three studies used only RT-PCR [23,27,30], one study used an unspecified throat swab [29], another study used an unspecified nasopharyngeal swab [28], and the remaining studies did not report the method for diagnosis [21,22,26]. All patients tested positive, except in one study [25] that enrolled two participants with negative RT-PCR test results reporting tinnitus and hearing loss. Of the patients that tested positive, eight patients were symptomatic with typical features of COVID-19, such as pneumonia, fever, and coughing; three patients were asymptomatic; and one patient’s symptoms were not described although she had no features of pneumonia. Only two studies reported follow-up testing, which determined a negative coronavirus using RT-PCR test and normal chest X-ray [24], and two negative coronavirus using respiratory swabs [30]. 

#### 3.4.9. Treatment of COVID-19

Treatment of COVID-19 varied among the studies. Six patients with varying degrees of COVID-19 symptoms were hospitalized, and management of symptoms involved medication, such as azithromycin, remdesivir, oseltamivir, and enoxaparin. Other treatments alleviating COVID-19 symptoms included high flow oxygen [22], intubation [26,27], and non-invasive mechanical ventilation [31]. Three studies reported conservative at-home treatment of coronavirus symptoms as one patient had no features of pneumonia [21], one patient did not require admission to the hospital [28], and another patient was given antiviral medication [24]. Management of COVID-19 was not described in the remaining studies.

#### 3.4.10. Quality Analysis of Case Reports

In total, three studies were of good quality, seven studies were of fair quality, and two studies were of poor quality (See Appendix A). There was a lack of follow-up assessments for seven case reports and outcome measures that were undefined or undeterminable in three studies. Despite the lack of details, most case reports were able to provide unbiased reports of audio-vestibular symptoms.

### 3.5. Cross-Sectional Studies Investigating Disease Impact 

#### 3.5.1. Study Characteristics

There were 17 clinical studies, including 8913 participants with an age range of 6 to 98 years. Some studies reported an equal gender divide, and others reported variable ratios, such as Munro et al. [44] reporting 88% and Viola et al. [47] reporting 67% of the participants were males, as seen in Table 3. One study included other genders [33] (e.g., nonbinary and cisgender), and another study did not report the prevalence of symptoms in males and females [32]. The number of patients included ranged from 6 to 1420 in these studies, with most being conducted in Europe followed by Asia. There were seven studies that took place in Europe from Italy [35,43,47], France [39,48], and England [44]. Lechien et al. [41] had participants from France, Italy, Spain, Belgium, and Switzerland. There were six studies from Asia reporting from India [45,46], Pakistan [36], Turkey [49,50,51], and China [42]. The three additional studies were located in Russia [32]; Egypt [37]; and internationally, including the USA, UK, Northern Ireland, France, Canada, Spain, Netherlands, Ireland, Sweden, and other countries [18,33]. 

#### 3.5.2. Study Designs and Outcomes

Studies were both retrospective (e.g., Elibol [34], Lechien et al. [41], Liang et al. [42], Klopfenstein et al. [39], and Zayet et al. [48]) and prospective observational studies (e.g., Daikhes et al. [32], Karadaş and Sonkaya [38], Özçelik Korkmaz et al. [40], and Swain and Pani [46]). Data collection was completed via verbal questioning during ear, nose, and throat (ENT) examinations for all of the included studies. Outcome measures included self-reported questionnaires within six studies [33,36,37,40,43,47]. Additionally, only one study used validated questionnaires (e.g., tinnitus handicap inventory (THI)) [35], and another study used a severity scale (i.e., visual analogue scale (VAS), ranging from 0 (absent) to 10 (most severe) [47]. Sources of heterogeneity included different tinnitus reporting criteria, age groups, and study focus. Two studies included control groups, namely, Daikhes et al. [32], who had 30 controls included, and Freni et al. [35], who had 20 controls with no history of hearing loss or tinnitus. Zayet et al. [48] included a control group who had Influenza and no COVID-19 symptoms. 

#### 3.5.3. Pre-Existing Health Conditions

Pre-existing health conditions were described in 11 studies that included hypertension, asthma, diabetes mellitus, cardiovascular disease, rheumatoid arthritis, arrhythmia, dyslipidemia, peptic ulcer, thyroid disease, musculoskeletal conditions, metabolic/endocrine conditions, neurological conditions, cancer, chronic obstructive pulmonary disease, gastroesophageal reflux disease, allergies, respiratory insufficiency/disease, chronic rhinosinusitis (CRS) with or without polyps, history of surgery for CRS, depression, allergic rhinitis, autoimmune diseases, chronic liver diseases/insufficiency, cerebrovascular disease, hyperlipidemia, anemia, renal failure/chronic kidney disease, sinonasal problems, hearing loss, tinnitus, vestibular disorders, immunosuppression, and other conditions not specified [34,35,36,37,39,40,41,42,43,44,48]. Some studies excluded participants with comorbidities such as patients with hearing loss or at risk of having a hearing loss (e.g., noise exposure, surgeries, ototoxic medication, or diseases that may lead to hearing loss) [32,46,47]. Other comorbidity exclusion criteria included psychiatric disorders, cardiovascular or circulatory disorders [47], treatment with new drugs, chronic nasal problems, recent head trauma, brain or nose operations, and severe respiratory failure [45]. Davis et al. [33] did not mention comorbidities; however, they excluded the following symptoms from the analysis: high blood pressure, low blood pressure, thrombosis, seizures, low oxygen levels, high blood sugar, and low blood sugar. 

#### 3.5.4. Tinnitus Overview

In total, 1763 participants reported tinnitus in the 17 included studies and an additional study by Beukes et al. [18]. This study did not directly investigate the COVID-19 disease but identified seven individuals reporting tinnitus and four with hearing loss after contracting COVID-19 from the sample of 237 reporting COVID-19 symptoms, out of the 3103 participants. Prevalence ranged from 0.35% [41] to 67% [45] for the disease related studies. The variability was found even in larger studies as Lechien et al. [41] had a prevalence of 0.35% for 1420 participants and Davis et al. [33] a prevalence of 34% for 3762 participants. Sensitivity analysis removing the outlier studies did not impact the results. The studies were not always clear if the tinnitus onset was post COVID or if it was tinnitus that was exacerbated. As heterogeneity was high (*I*^2^ = 97.91, *p* < 0.001), a random-effect meta-analyses was conducted. The pooled prevalence estimate (Figure 2) for tinnitus associated with COVID-19 from these 17 cross-sectional studies and the Beukes et al. [18] study (18 studies) was 8% (CI: 5 to 13%).

#### 3.5.5. Tinnitus Characteristic

As no standard standardized diagnostic criterion for tinnitus was used, great variability was found regarding tinnitus severity and characteristics, and not all studies described the tinnitus. Viola et al. [47] presented tinnitus descriptions for participants to select, indicating large variability in the tinnitus experienced. Amongst 43 patients, 17 (39.5%) described tinnitus as recurrent (comes and goes away during the day), 10 (23.3%) as occasional (episodic, sporadic), 7 (16.3%) as continuous fluctuating with intensity changes throughout the day, 4 (9.3%) as persistent (always present, day and night), 3 (7.0%) as pulsatile (synchronous with heartbeat), and 2 (4.6%) as continuous (always present with the same intensity, making it difficult to fall asleep). VAS mean score for tinnitus was 5, revealing an overall moderate severity across patients [47]. Freni et al. [35] reported a THI score of 6.6 ± 12.1 (THI scores of 0–16 are considered as no or slight handicap) and that for 10 patients tinnitus was initiated or worsened due to COVID-19. In a study focused on the pandemic impact [18], among those with pre-existing tinnitus who contracted COVID-19 (*n* = 237), 40% reported that their tinnitus became more bothersome, 54% reported no changes to their tinnitus, and 6% reported improvement in their tinnitus. Those reporting an improvement mentioned that they had gained new perspectives and realized that their tinnitus was not such a big problem compared with fighting to survive while hospitalized with COVID-19. For those reporting their tinnitus worsened, it is unclear whether reported changes were directly related to the virus or not. Other factors may have played a role, for instance, participants taking medications or vitamins to boost the immune response reported a significant increase in their tinnitus.

*Tinnitus location:* One patient reported unilateral tinnitus (lateralized left) associated with aural pressure among eight identified self-reports of tinnitus [44], and the tinnitus location was not reported in other studies. 

*Tinnitus onset:* Tinnitus onset was reported from one day post-infection [40,42] and 1-week post-infection by 11.5% (10.5%–12.5%) in the study by Davis et al. [33], increasing to 26.2% (23.5%-29.1%) over 6–7 months post-COVID-19. Davis et al. [33] identified that tinnitus was one of the later symptoms to occur at approximately 7 weeks post-COVID-19. 

*Tinnitus duration:* Few studies mentioned tinnitus duration, and where reported, great variation was found. In the study by Munro et al. [44], there were eight individuals with tinnitus, of whom three also reported a pre-existing hearing loss. Of these, one participant reported that the tinnitus resolved over time. Savtale et al. [45] revealed 120 patients (66.66%) amongst 188 self-reported new-onset tinnitus lasting 5 days (median, interquartile range [IQR] 4–6). Liang et al. [42] revealed the average duration of tinnitus was 5 days. Özçelik Korkmaz et al. [40] revealed duration ranging from 1 to 9 days (median = 4). Davis et al. [33] reported that both tinnitus and hearing loss were likely to ramp up sharply in the first two months and continue to increase up to 6–7 months post COVID-19. 

*Tinnitus management:* Management of tinnitus was not described in any of the studies. This may be due to the unknown etiology between coronavirus and tinnitus and the inconsistency in defining and reporting tinnitus, leading to variability in estimates. 

#### 3.5.6. Hearing Loss

Of the 16 included cross-sectional studies, 10 also examined hearing loss as a possible symptom of COVID-19 (*n* = 495), although there was substantial variability in how studies assessed and reported hearing loss. Gender and age were, for instance, not reported in most studies except Swain and Pani [46] and Özçelik Korkmaz et al. [40]. Study designs included retrospective evaluation of medical records for 5 studies [32,34,39,46,48]; verbal questionnaire interviews [44,45]; and the use of self-reported symptoms questionnaires for 2 studies [33,40]. Only one study reported Hearing Handicap Inventory for Adults (HHIA) scores for all patients [35]. 

The overall prevalence figures ranged from 0% [28] to 100% [45]. Swain and Pani [46] identified 28 patients ranging from 16 years to 52 years (mean = 28.2), with 15 (53.57%) females and 13 (46.42%) males with hearing loss after hospital discharge. When grouped by age, Özçelik Korkmaz et al. [40] found hearing loss prevalence was 50% (*n* = 3) for patients 60 years and older and 50% (*n* = 3) for those younger than 60 years, with two males and four females. 

*Type of hearing loss:* Where provided, hearing loss ranged from being mild to moderate in degree [45], high frequency in pattern [35,46], conductive [46], and sensorineural [45,46]. Although bilateral sensorineural hearing loss was identified, the majority included unilateral hearing loss (e.g., 83% of the sample by Munro et al. [44]). 

*Onset and progression of hearing loss:* Davis et al. [33] revealed that the incidence of hearing loss increased from 2.98% (CI: 2.47–3.54%) in week 1 to 6.42% (CI: 5.00–8.07%) of respondents in week 6–7. Another study reporting duration found hearing impairment lasting from 3 to 7 days (median = 4) [40], and Savtale et al. [45] identified self-reported new onset hearing loss lasting 13 days (median, 9.5–16.75 IQR). Freni et al. [35] revealed the appearance or worsening of hearing loss in 20 patients (40%), with an HHIA score of 13.2 ± 14.9 during the active phase of symptomatology from COVID-19. After recovery (15 days after negative RT-PCR test), 9 patients reported the presence of hearing loss, with a lower total mean of the HHIA score of 4.24 ± 5.55 (*p* < 0.001).

*Assessment and management of hearing loss:* Most studies relied on self-reports of hearing loss, and only one study undertook a full audiological evaluation consisting of tympanometry, acoustic reflex thresholds, and transient evoked otoacoustic emissions (TEOAEs) [32]. Pure tone audiometry was conducted in two studies [35,46], tympanometry in one study [46], and TEOAEs in three studies [32,35,46]. One study utilized a tuning fork test at a frequency of 512 Hz to examine audiologic function [45]. TEOAE amplitude was significantly worse in 22/28 COVID positive cases [46] and was also worse compared to individuals without COVID-19 [32,35]. Only one study among the others reported treatment for sudden sensorineural hearing loss (SSNHL) using corticosteroid therapy, specifically oral prednisolone for three weeks along with vitamin B-complex and proton pump inhibitor daily [46].

#### 3.5.7. Vestibular Deficits

Reports of vestibular impairments were found in four clinical studies as a possible symptom of COVID-19. All studies assessed self-reported questionnaires regarding otologic symptoms of COVID-19. Among the studies, only one utilized a severity scale to investigate the severity of balance disorders [47]. Davis et al. [33] reported that dizziness/balance issues were most likely to persist after six months. In this study, 30–50% of the respondents experienced dizziness/balance issues after six months. Özçelik Korkmaz et al. [40] reported that two participants had a previous vestibular disorder, with 31.8% of the participants having dizziness and 6% having true vertigo post-COVID-19. Dizziness was statistically significantly higher in women that were less than 60 years old, and true vertigo was only present in participants younger than 60 years old. The range of duration for true vertigo for participants was 1 to 5 days with a median of 3 days. For the duration of dizziness, the range was 2 to 13 days with a median of 6 days. Micarelli et al. [43] stated that 6.2% of the participants experienced vertigo/dizziness and 6.3% of the participants experienced disequilibrium. Vertigo or dizziness symptoms had duration ranges of 2 to 12 days, while disequilibrium was 2 to 14 days. In Viola et al. [47], 18.4% of the participants reported balance disorders after the diagnosis of COVID-19. Of those with balance deficits, 94.1% experienced dizziness and 5.9% experienced acute vertigo attacks. Fourteen (7.6%) had both tinnitus and an equilibrium disorder, while 7% experienced a migraine and an equilibrium disorder. There were 20 (58.8%) females and 14 (41.2%) males that experienced balance deficits. The severity of the equilibrium disorders was measured by the VAS. The mean score for the equilibrium disorders was 5 out of a 1–10 rating. Management or treatment of vestibular impairments was not discussed in the studies. 

#### 3.5.8. COVID-19 Testing

All the studies used RT-PCR testing for the diagnosis of COVID-19 as part of the inclusion criteria, except for Viola et al. [47], who used an unspecified nasopharyngeal swab. Davis et al. [33] and Macarelli et al. [43] included those who had experienced COVID-19 symptoms but not been tested. Micarelli et al. [43] also required the participants to have no fever in the past 14 days or a negative test for COVID-19 to participate in the study. Iqbal et al. [36] and Kamal et al. [37] only included participants who had PCR testing, to evaluate the presence or absence of SARS-CoV-2. 

#### 3.5.9. Treatment of COVID-19

Treatment of COVID-19 was not always described, and some studies only stated that patients were hospitalized [33,34,35,40,44,45,46,47] or in intensive care units [36,37,39,48]. In Daikhes et al. [32], groups of drugs were used as treatments (antiviral, antimalarial, anticoagulants, and antibacterial). Lechien et al. [41] used oral treatment depending on symptoms such as analgesic drugs (paracetamol, nonsteroidal anti-inflammatory, oral corticosteroids, mucolytics, hydroxychloroquine), antibiotics (macrolides), and beta-lactam antibiotics, along with antiviral drugs, pulmonary aerosols, and nasal treatments. Other treatments for COVID-19 symptoms included oxygen therapy [36,39,48], home remedies [36], and vitamins [37]. Liang et al. [42] stated that the treatment guidelines for COVID-19 issued by the National Health Committee of the People’s Republic of China were used for treatment. Micarelli et al. [43] did not provide information regarding the treatment of COVID-19. Furthermore, many of the clinical studies had patients who did not receive treatment.

#### 3.5.10. Quality Analysis of Cross-Sectional Observational Studies

Three studies were of good quality, seven studies were of fair quality, and no studies were of poor quality (Appendix A). 

### 3.6. Pandemic Impact Studies: Comparing Tinnitus before and during the Pandemic 

#### 3.6.1. Study Characteristics

There were three studies comparing tinnitus severity before and during the pandemic performed in Italy [50], Germany [51], and China [52], as summarized in Table 4. The number of participants varied (16, 94, and 122, respectively). These studies focused on how the COVID-19 pandemic affected tinnitus rather than on how the actual COVID-19 virus affected tinnitus. Therefore, COVID-19 testing (positive or negative) information was not included in the data collection for these studies. The THI questionnaire was used as part of the assessment of tinnitus severity for all three studies. As Anzivino et al.’s study [50] was a letter to the editor, the study was not detailed in terms of describing the age and gender characteristics of participants. In Schlee et al. [51] and Xia et al. [52], the mean age was similar, 54.0 (SD: 10.9) and 52.6 (SD: 14.7), respectively; however, regarding gender percentages, the male percentage in the participants was greater in Schlee et al. [51], at 65.5% and 48.3%, respectively.

#### 3.6.2. Tinnitus Characteristics

Overall, the studies showed there was an increase in tinnitus severity during the pandemic. Anzivino et al. [50] found that the grade of tinnitus severity had increased by one level on the THI for a small sample tested (12 out of 16 participants) during the pandemic. Schlee et al. [51] found that although there was an increase in tinnitus severity on 122 patients during the pandemic compared with before, as measured by the THI and Mini-Tinnitus Questionnaire (Mini-TQ), this difference was not significant. Tinnitus severity was, however, significantly correlated to pandemic-related stress using the social isolation electronic survey to identify grief, frustration, stress, and nervousness. The study also revealed that the higher the participant’s neuroticism score, the more distinct was the worsening of the tinnitus. Xia et al. [52] identified significantly higher tinnitus severity during the pandemic (40 out of 100 for the THI for 99 patients) compared to before the pandemic (34 out of 100 for the THI for 89 patients) and that the effect of anxiety (measured by Self-rating Anxiety Scale; SAS) associated with the impact of the pandemic appeared to contribute to elevated tinnitus awareness. 

#### 3.6.3. Tinnitus Treatments

Xia et al. [52] reported that educational counselling resulted in improvements in the SAS, THI score, and tinnitus loudness test before the pandemic, but such treatments were less effective in 2020. The authors concluded that educational counselling was not enough for the stress and anxiety during the COVID-19 pandemic and provided evidence that anxiety is a contributing factor to tinnitus severity.

#### 3.6.4. Quality Analysis of Pandemic Impact Study Comparing Tinnitus before and during the Pandemic

Anzivino et al.’s [50] study was rated as poor due to the lack of description of the participants (e.g., gender, age, and eligibility criteria), the lack of statistical analysis, lack of repeated outcomes measures, and the small sample size (*n* = 16). Schlee et al. [51] and Xia et al. [52] were rated as fair due to providing a relatively good description of the aim, eligibility criteria, outcome measures, and fair sample size.

### 3.7. Cross-Sectional Studies Investigating the Effects of the Pandemic on Pre-existing Tinnitus 

#### 3.7.1. Study Characteristics

There were two cross-sectional studies (Table 4) investigating the effect of the pandemic on pre-existing tinnitus at one point in time [18,49]. Although these studies tried to identify the incidence of tinnitus during the pandemic, most of the included participants had pre-existing tinnitus. Drawing conclusions regarding the impact of the pandemic on the incidence of tinnitus compared with the incidence before the pandemic can thus not be determined.

#### 3.7.2. Outcome Measures

Beukes et al. [18] used the tinnitus handicap inventory screening (THI-S) versions to measure the severity of tinnitus, as well as an online survey that contained questions regarding demographics, contracting COVID-19, whether social distancing guidelines were followed, the emotional and financial toll of the pandemic, and the use of coping strategies. Naylor et al. [49] assessed the impact of the pandemic on those with hearing loss using an online survey that asked questions regarding behavior, emotions, hearing performance, practical problems (wearing hearing aids and masks), and tinnitus during the pandemic. 

#### 3.7.3. Individual Study Descriptions

To study a more heterogeneous tinnitus population, Beukes et al. [18] surveyed 3103 individuals with tinnitus between May-June 2020. Although global representation was sought, the majority of the participants were from North America (49%) and Europe (47%), with a minority (4%) representing other world regions and a total number of 3103 participants equally balanced in gender. Findings indicated that the pandemic had not altered tinnitus for the majority (67%), 31% reported their tinnitus was exacerbated during the pandemic, and 2% found their tinnitus was better. Tinnitus was found to be significantly more bothersome during the pandemic for females and younger adults under the age of 50. Additional mediating factors significantly exacerbating tinnitus included self-isolating, experiencing loneliness, sleeping poorly, and reduced levels of exercise. Increased depression, anxiety, irritability, and financial worries further significantly contributed to tinnitus being more bothersome during the pandemic period. Participants from the Naylor et al.’s [49] study consisted of 129 individuals with hearing loss that lived in Glasgow, Scotland. Ages ranged from 27 to 76 (mean = 64.4) years old with 48% female. Data were collected from 29 May to 15 June 2020; therefore, the participants had experienced over 2 months of lockdown. Due to the focus on hearing loss, there was only one question about tinnitus in the online survey for Naylor et al. [49]; the primary outcome of the study was to determine the impact of the pandemic on those with hearing loss. Participants were grouped into those with worse hearing (*n* = 61) and better hearing (*n* = 68). Out of the 129 participants, 70 had pre-existing tinnitus. In the worst hearing group, 38 had tinnitus while the better hearing group had 32 participants with tinnitus. In response to the statement in the survey, “My tinnitus has been worse since lockdown started,” 42.1% agreed, 31.6% were neutral, and 26.3% disagreed in the worse hearing group. However, in the better hearing group, 18.8% agreed, 37.5% were neutral, and 43.8% disagreed. There was a non-significant trend toward tinnitus being worse during the pandemic for those with greater hearing loss. Participants explained that tinnitus was more noticeable when the world around them was quieter. Overall, the studies showed that there may be a trend for tinnitus to exacerbate during the pandemic; however, this did not pertain to the majority of participants with pre-existing tinnitus. Contributing factors may include gender, age, self-isolation, loneliness, lack of sleep and exercise, depression, anxiety, irritability, financial concerns, or a quieter environment. 

#### 3.7.4. Quality Analysis of Cross-Sectional Studies Investigating Effects of the Pandemic on Pre-Existing Tinnitus

The quality analyses of these two cross-sectional studies were both rated as fair, as seen in Table 4. Guidelines for consistency of reporting in future COVID-19 studies are provided in Table 5 to help science progress and improve patient outcomes going forward. 

## 4. Discussion

The purpose of this systematic review was to review the evidence regarding the effect of contracting COVID-19 on tinnitus and the effect of the wider pandemic on tinnitus. Until and including 31 March 2021, there were 33 published articles discussing these effects. These studies varied in study design and purpose. There were 28 investigating the impact of COVID-19 on tinnitus and five reporting the impact of the pandemic on tinnitus. Although not all countries published reports, there was a fair global representation, including two studies that attempted international data collection. This discussion highlights the main findings. 

### 4.1. The Effect of Contracting COVID-19 on Tinnitus 

No consistent profile regarding who may develop tinnitus post-COVID-19 was identified. A range of ages was affected (6 to 98 years) and there were variations in gender proportions, possibly attributed to different research designs. From this review, no consistent pattern was identified regarding the risk of developing tinnitus. Some individuals had pre-existing conditions such as head trauma, asthma, diabetes, hypertension, cancer, and hearing disorders, but others had no pre-existing comorbidities. COVID-19 factors that may have contributed were furthermore unclear as not all individuals were tested for the presence of COVID-19, and some studies relied on self-reporting. When reported, the RT-PCR test was most frequently used. The severity of the COVID-19 symptoms also varied, resulting in some individuals being hospitalized, ventilated, and medicated, while others remained at home. It is not clear from any studies as to whether the severity of the infection or treatment provided for COVID-19 correlated with the tinnitus severity, presentation, or duration. From 17 included studies, the estimated prevalence was 8% (CI: 5 to 13%). This is between the prevalence rates reported by Almufarrij and Munro [10] of 14.8% (CI: 6.3 to 26.1) and Jafari et al. [11] of 4.5% (CI: 1.2 to 15.3). All three reviews included different studies due to different protocols followed but provide some insights into the possible expected prevalence from the published literature. 

The onset of the tinnitus post-COVID-19 was variable. This included reports of onset between 1 day [42] and 7 weeks post-onset. Interestingly, Davis et al. [33] reported that the incidence increased from 11.5% at 1-week post-infection to 26.2% by week 6–7 post-COVID-19 and that tinnitus was one of the later symptoms to develop. Due to the prevalence of tinnitus being at least 11% [58] within the general adult population, it is difficult to determine if other factors may have contributed to the tinnitus experienced. Tinnitus duration also varied, with some reporting a resolution after 5 days (e.g., Liang et al. [42], Özçelik Korkmaz et al. [40], and Savtale et al. [45]) and Davis et al. [33] reporting tinnitus to increase in later months post COVID-19. Due to the sudden and rapid developments of COVID-19, there was not always the option for large-scaled studies due to the time pressures, and most studies were retrospective or observational cross-sectional studies. A lack of longitudinal tracking regarding the progression of tinnitus was not always incorporated, hence longer-term trajectories or the tinnitus presentations were not identified. 

Tinnitus presentations were often not provided. Viola et al. [47] found that tinnitus was more frequently recurrent and occasional as opposed to persistent and continuous, but only 43 individuals were included in this study. The tinnitus location varied between unilateral and bilateral presentations, although the location was often not described. It was not always clear whether there was pre-existing tinnitus. Beukes et al. [18] found that of those with pre-existing tinnitus who contracted COVID-19 (*n* = 237), 40% reported that their tinnitus became more bothersome, 54% reported no changes to their tinnitus, and 6% reported improvement in their tinnitus, again indicating inconsistent consequences of COVID-19 on tinnitus. 

### 4.2. Characteristics of Other Auditory Vestibular Conditions 

Although not always reported, various individuals presented with both tinnitus and hearing loss, with unilateral SNHL being most commonly reported (e.g., Munro et al. [44]). Variability in the hearing loss severity was also found with ranges between mild to severe. Some studies reported that the hearing recovered [40,56], and others found that it deteriorated between 1 to 7 weeks post-COVID-19 [33]. Dizziness and vertigo were also reported, although the prevalence was lower. These auditory symptoms were reported to resolve by some and persist 6–7 months post COVID-19 by others (e.g., Davis et al. [33]). Studies identifying the mechanisms and associations of these symptoms with COVID-19 as well as the trajectory of these symptoms are required. 

Although speculative, numerous pathogenesis have been proposed regarding the possible association between hearing loss and the SARS-CoV-2 virus. Findings by Daikhes et al. [32], Freni et al. [35], and Swain and Pani [46] regarding reduced TEOAE amplitudes have been supported by Mustafa [59], who found that high-frequency pure-tone thresholds and TEOAE amplitudes were significantly worse in 20 asymptomatic COVID-19 PCR-positive cases when compared with 20 normal non-infected participants. This indicates that the SARS-CoV-2 virus could affect cochlear outer hair cell functioning. Further suggested mechanisms suggest that the SARS-CoV-2 infection together with serotonin release and blood coagulation may intertwine to activate platelets and drive SSNHL [60]. Excessive cytokine release and/or ischemic damage from thrombosis are furthermore suggested to increase oxidative damage, resulting in permanent hearing damage [61].

### 4.3. The Impact of the Pandemic on Tinnitus Experiences 

A wide range of individuals reporting tinnitus during the pandemic was identified with variations in gender divides. Overall, these studies found that tinnitus severity often increased during the pandemic but not for all individuals. Stress, neuroticism, and anxiety were identified as contributing factors [51,52]. Beukes et al. [18] found that tinnitus was significantly more bothersome during the pandemic for females and younger adults under the age of 50. Additional mediating factors that significantly exacerbated tinnitus included self-isolating, experiencing loneliness, sleeping poorly, and reduced levels of exercise. Increased depression, anxiety, irritability, and financial worries further significantly contributed to tinnitus being more bothersome during the pandemic period. According to these studies, there is a correlation between the emotional toll of the pandemic and the severity of tinnitus in participants; however, there is a need for in-depth studies to determine certain factors contributing to the elevated tinnitus severity and what therapy or tools can be provided to counteract these factors. As these studies have generally included individuals who had pre-existing tinnitus, conclusions regarding the impact of the pandemic on the incidence of tinnitus compared with the incidence prior to the pandemic cannot be drawn. 

### 4.4. Limitations of the Evidence and Review Process 

Although this review aimed to conclude the presentation of tinnitus, these descriptions were generally not provided. This made concluding the risk factors, tinnitus characteristics, progression of the tinnitus, and other audio-vestibular deficits. Due to the variations in what was reported and how tinnitus and or audio-vestibular difficulties were measured, it made the synthesis incomplete. The results presented are limited due to variability in study design and approach as well as inconsistent use of outcome measures. Follow-up reporting was also poor. Clear descriptions of tinnitus were not provided in all the studies, making synthesis of the studies difficult. Only a few of the studies for instance specifically described the tinnitus or investigated the onset, duration, severity, characteristics, and psychological impact thereof. Although overall the study quality was fair and represented unbiased reports, quality was compromised as all but three studies had no control or comparator group. Furthermore, self-reported assessment measures were generally included, relying on participant’s recall of symptoms and progression. Further factors of bias included questionnaire distribution through tinnitus associations, which could furthermore inflate pooled estimates of tinnitus or only participants from one region. 

### 4.5. Implications for Practice, Policy, and Future Research 

These findings have important implications for clinical services. As identified by Almufarrij and Munro [10], tinnitus is the most prevalent audio-vestibular symptom (14.8%) post COVID-19. Health professionals who may be involved with COVID-19 patients should be mindful that contacting COVID-19 may lead to tinnitus and other audiovestibular difficulties and such individuals should be directed to appropriate care. The COVID-19 pandemic undoubtedly disrupted and transformed usual healthcare services. Raising greater awareness among healthcare providers is required, due to the impact the COVID-19 virus and wider pandemic factors have on tinnitus and other audiological conditions. Despite studies identifying bothersome tinnitus, most did not discuss how tinnitus was managed. Xia et al. [52] mentioned that educational counselling that was normally helpful was not as effective for those with bothersome tinnitus during the pandemic. They put this down to needing management strategies that addressed anxiety and the increased stress during the pandemic. Those presenting with bothersome tinnitus during the pandemic or post COVID-19 may thus require different tinnitus management approaches. 

Those with tinnitus often mention that they are unsupported as healthcare professionals are not understanding of the difficulties they face due to tinnitus and hearing loss [62,63]. Educating healthcare professionals specifically question experiences of any such symptoms, so that these individuals can be directed to the most appropriate care. Patient associations and audiologists should also be available to reassure and help those now experiencing tinnitus or with more bothersome tinnitus. Specific needs of those with tinnitus identified during the COVID-19 pandemic can be used by healthcare providers to shape future tinnitus services. These include a wider range of support for tinnitus and hearing-related difficulties, including more affordable hearing healthcare such as hearing aids and hearing protection. Those with tinnitus furthermore desire means of social support and education to the general population regarding the impact of tinnitus [62,63]. They also indicated the need for support to better deal with the increased stress and anxiety related to the pandemic. Individuals with tinnitus indicated that tinnitus-related research should be prioritized, including searching for tinnitus cures. Overall, there is a need for (a) understanding professional support and access to multidisciplinary experts, (b) a greater range of therapies and resources, (c) access to more information about tinnitus, (d) prioritizing tinnitus research, and (e) more support for hearing protection and hearing loss prevention. Patient organizations and professionals should be encouraged to work together to provide improved outlets for tinnitus care. Most importantly, digital therapeutical approaches should be prioritized to provide psychological interventions to those suffering from tinnitus and not able to access services due to demand on healthcare as well as not having access to services such as these, which are seen as low priority by hearing healthcare professionals during the pandemic [64]. Several studies across the globe have demonstrated the efficacy and effectiveness of Internet-based cognitive behavioral therapy (ICBT) for tinnitus [65,66,67,68], although not many programs are available for individuals. For this reason, clinicians and policymakers need to consider alternative ways of offering tinnitus services using teleaudiology approaches. 

The wider pandemic effect, such as the impact of the use of non-transparent face masks hampering lip-reading and face coverings reducing the acoustic transmission, attenuating the sound, and preventing lip reading, makes it difficult for those with auditory difficulties, especially those with greater difficulties, e.g., cochlear implant users. A study of 59 patients with hearing loss attending hospital appointments in Italy indicated that 37% reported moderate and 24% severe hearing difficulties [69]. These difficulties may contribute to the reports of increased anxiety during the pandemic for individuals with hearing loss, as demonstrated by a study focusing on 56 Iranian hard of hearing and deaf adolescence [70]. Support for those with hearing loss and other auditory symptoms is thus required. 

While the current literature provides some early understanding of the link between COVID-19 and tinnitus, due to limitations in terms of study design as well as issues with reporting of study findings, the conclusions drawn from this review are preliminary. 

## 5. Conclusions

This review has been helpful in identifying the impact of both COVID-19 and the pandemic on tinnitus. Findings were limited to the quality of the research presented. This review identified a need for consistency in reporting and gathering data to be able to synthesis information. This review provides a foundation on which further robust research can be designed. What is important is investigating the mechanisms of these changes. It is not known if tinnitus and hearing loss can be directly attributed to the COVID-19 virus or whether they are attributed to other factors. These may include the impact of receiving critical care, including ototoxic medications [71], especially for those with a possible greater vulnerability to ototoxicity [72]. The precise pathophysiological mechanisms causing tinnitus and other auditory-related symptoms remain unclear, and more research is required to further investigate these mechanisms. 

## Figures and Tables

**Figure 1 jcm-10-02763-f001:**
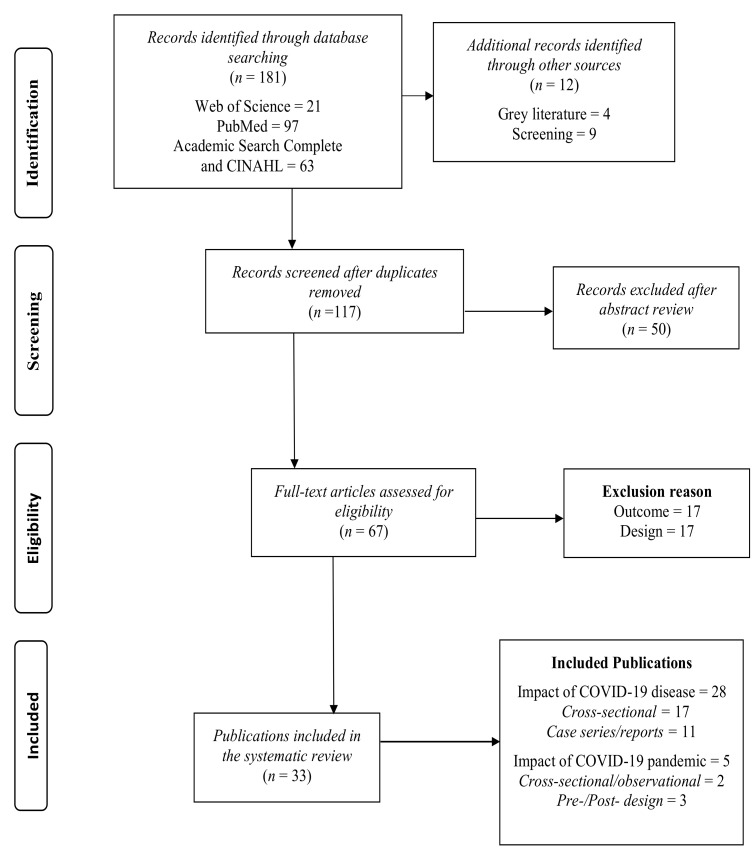
Prisma Flowchart.

**Figure 2 jcm-10-02763-f002:**
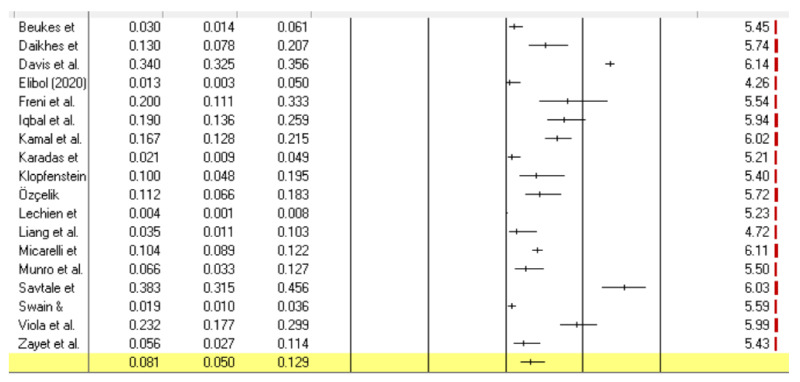
Forest plot for the estimated prevalence of tinnitus in suspected and probable COVID-19 cases. The individual study estimate, and the 95% CI are represented by center lines and their error bars, respectively.

**Table 1 jcm-10-02763-t001:** Inclusion and exclusion criteria for the review.

	Inclusion	Exclusion
Population	Individuals of any age experiencing tinnitus during the COVID-19 pandemic or due to contracting SARS-CoV-2 (COVID-19).	Individuals without tinnitus
Condition	Tinnitus, which is the perception of sound in the ears or head in the absence of any external sound.	No tinnitus
Exposure	Confirmed, probable, or suspected exposure to COVID-19 or the pandemic.	Not exposed to COVID-19 or the pandemic.
Comparator	Not applicable	Not applicable
Outcomes	Self-reported experiences of tinnitus	No tinnitus reports
Study designs	Any study designs, including commentaries and case studies	Systematic reviews, secondary studies discussing other studies
Timings	At least one time point	No exclusions regarding the length of follow up assessments
Language	All languages	None

**Table 2 jcm-10-02763-t002:** Summary of the included COVID-19 disease impact case reports/case series studies.

Study	Country Study Period	Quality Assessment	Participant Characteristics					Other Audio-Vestibular Manifestations			
			Publication Date	Participants (*n* = 35)	Gender	Age in Years (Mean, Median, Range)	Tinnitus	Hearing Loss	Vestibular Impairments	Taste Disorders	Smell Disorders
Chirakkal et al. [21]	State of Qatar Not stated	Fair	04/12/2020	*n* = 1	M = 0% F = 100%	35	*n* = 1 (Ringing, matched at 4 kHz at 10 dBHL)	X			
Cui et al. [22]	China 14/01/2020–20/30/2020	Fair	01/07/2020	*n* = 20	M = 55% F = 45%	NA, 63 32–72	*n* = 1		X		X
Degen et al. [23]	Germany Not stated	Fair	01/08/2020	*n* = 1	M = 100% F = 0%	60	*n* = 1 (bilateral, described as loud white noise)	X			
Fidan [24]	Turkey Not stated	Fair	01/05/2020	*n* = 1	M = 0% F = 100%	35	*n* = 1	X			
Karimi-Galougahi et al. [25]	Iran Not stated	Fair	10/06/2020	*n* = 6	M = 33% F = 67%	32	*n* = 4	X	X		
Koumpa et al. [26]	United Kingdom Not stated	Good	13/10/2020	*n* = 1	M =100% F = 0%	45	*n* = 1 (left sided)	X			
Lamounier et al. [27]	Brazil 12/03/2020–23/05/2020	Good	03/11/2020	*n* = 1	M = 0% F = 100%	67	*n* = 1 (disabling)	X			
Lang et al. [28]	Ireland 19/04/2020–09/06/2020	Fair	01/10/2020	*n* = 1	M = 0% F = 100%	30	*n* = 1 (right-sided)	X			
Maharaj & Hari [29]	Malaysia Not stated	Good	23/10/2020	*n* = 1	M = 100% F = 0%	44	*n* = 1 “Right-sided non-pulsatile”		X		
Abdel-Rhman and Abdel- Wahid [30]	Egypt 15/04/2020–05/2020	Fair	08/07/2020	*n* = 1	M = 100% F = 0%	52	*n* = 1 Gradually worsening	X			
Sun et al. [31]	China 16/01/20202–24/02/2020	Poor	01/05/2020	*n* = 1	M = 100% F = 0%	38	*n* = 1	X			

**Table 3 jcm-10-02763-t003:** Summary of the included COVID-19 cross-sectional studies.

Study	Country Study Period	Quality Assessment	Participant Characteristics					Other Audio-Vestibular Manifestations			
			Publicat-ion Date	Participants (*n* = 8913)	Gender	Age in Years (Mean, Median, and Range)	Tinnitus	Hearing Loss	Vestibular Impairments	Taste Disorders	Smell Disorders
Daikhes et al. [32]	Russia 4/2020–6/2020	Poor	20/07/2020	*n* = 108 (including *n* = 30 as a control)	Not provided	NA, 20–50, NA	*n* = 14 (17%)	X			
Davis et al. [33]	International (39 countries) 9/6/2020–11/25/2020	Fair	26/12/2020	*n* = 3762	M = 19.1% F = 78.9% Other = 2%	NA, NA,30–60	*n* = 1280 (34%)	X	X	X	X
Elibol [34]	Turkey 3/25/2020–4/25/2020	Fair	01/09/2020	*n* = 155	M = 41.3% F = 58.7%	36.3 (8.1), NA, 18–72	*n* = 2 (1.3%)	X		X	X
Freni et al. [35]	Italy Not stated	Fair	18/06/2020	*n* = 50 (including *n* = 20 as a control)	M = 60% F = 40%	37.7 (17.9), NA, 18–65	*n* = 10 (20%)	X		X	X
Iqbal et al. [36]	Pakistan 9/2020–12/2020	Good	02/02/2021	*n* = 158	M = 44.9% F = 55.1%	32.1 (12.42), NA, 19–80	*n* = 30 (19%)			X	X
Kamal et al. [37]	Egypt Not stated	Fair	29/09/2020	*n* = 287	M = 35.9% F = 64.1%	32.3 (8.5), NA, 20–60	*n* = 48 (17%)				
Karadaş et al. [38]	Turkey	Fair	25/06/2020	*n* = 239	M = 55.6% F= 44.4%	46.46 (15.41), 19–88	*n* = 5 (2.1%)	X	X	X	X
Klopfenstein et al. [39]	France 3/1/2020–3/14/2020	Fair	04/08/2020	*n* = 70	M = 33% F = 67%	47 (16), NA, NA	*n* = 7 (10%)	X		X	X
Özçelik Korkmaz et al. [40]	Turkey 4/2020–5/2020	Fair	03/10/2020	*n* = 116	M = 50% F = 50%	57.4 (14.32), NA, 19–83	*n* = 13 (11%)	X	X	X	X
Lechien et al. [41]	Europe 3/22/2020–4/10/2020	Fair	30/04/2020	*n* = 1420	M = 32.3% F = 67.7%	39.17 (12.09), 37, NA	*n* = 5 (0.3%)			X	X
Liang et al. [42]	China 3/16/2020–4/12/2020	Fair	24/06/2020	*n* = 86	M = 51.2% F = 48.8%	NA, 25.5, 6–57	*n* = 3 (3.5%)			X	X
Micarelli et al. [43]	Italy 3/23/2020–3/30/2020	Fair	20-Oct	*n* = 1380	M = 39.3% F = 60.6%	NA, 23–72, NA	*n* = 144 (10%)		X	X	X
Munro et al. [44]	UK Not stated	Fair	31/07/2020	*n* = 121	M = 87.5% F = 12.5% (of those with a change in tinnitus/hearing)	NA, 64, 44–82	*n* = 8 (7%)	X			
Savtale et al. [45]	India 10/1/2020–10/15/2020	Fair	08/02/2021	*n*= 180	M= 33.4% F= 66.6%	37.8 (12.5), NA, 18–65	*n* = 120 (67%)	X		X	X
Swain andPani [46]	India 3/2020–8/2020	Fair	03/02/2021	*n* = 472	M = 64.3% F = 35.7%	28.2, NA, 16–52	*n* = 9 (2%)	X		X	X
Viola et al. [47]	Italy 5/5/2020–6/10/2020	Fair	23/10/2020	*n* = 185	M = 53.5% F = 46.5%	52.15 (13), 53, 19–81	*n* = 43 (23%)		X		
Zayet et al. [48]	UK (France) 2/26/2020–3/14/2020	Fair	16/06/2020	*n* = 124, COVID group: *n* = 70; Influenza group: *n* = 54	Overall group: M = 31% F = 69% COVID only group: M = 41.4% F = 58.6%	59(13), NA, 19–98	*n* = 7 (6%) out of the COVID only group	X		X	X

**Table 4 jcm-10-02763-t004:** Summary of the included COVID-19 pandemic impact studies.

Study	Country Study Period	Quality Assessment	Participant Characteristics					Other Audio-Vestibular Manifestations		
			Publication Date	Participants (*n* = 3558)	Gender	Age in Years (Mean, Median, and Range)	Tinnitus	Hearing Loss	Vestibular Impairments	Taste/Smell Disorders
**Cross-sectional studies (*n* = 3232)**										
Beukes et al. [18]	International 4/29/2020–6/21/2020	Fair	05/11/2020	*n* = 3103	M = 50%F = 50%	58 (14), NA, 18–100	*n* = 3996 had pre-existing tinnitus, *n* = 7 post-COVID tinnitus (0.2%)	X		
Naylor et al. [49]	Scotland 5/29/2020–6/15/2020	Fair	01/11/2020	*n* = 129	M = 51.9% F = 48.1%	64.4, NA, 27–76	*n* = 70	X		
**Pre-/post-design (*n* = 326)**										
Anzivino et al. [50]	Italy ~5/1 to 5/15/2020	Poor	22/06/2020	*n* = 16	Not provided	Not provided	*n* = 16			
Schlee et al. [51]	Germany 3/28/2018–8/20/2018 and 4/14/2020–4/29/2020	Fair	26/08/2020	*n* = 122	M = 65.6% F = 34.4%	54.0 (10.9), NA, NA	*n* = 122	X	X	
Xia et al. [52]	China 3/1/2019–4/14/2019 and 3/1/2020–4/14/2020	Fair	05/02/2021	*n* = 188, *n* = 89 prior the pandemic, *n* = 99 during pandemic	2020:M = 43.4% F = 56.6% 2019: M = 48.3% F= 51.7%	2020 = 50.8 (15.1), NA, NA, 2019 = 52.6 (14.7), NA, NA	*n* = 188	X		

**Table 5 jcm-10-02763-t005:** Recommendations for future research investigating the impact of COVID-19 on audio-vestibular conditions.

Study Design Considerations	Data Collection and Reporting Suggestions
Including control groups with and without the presence of the disease or symptoms being investigated	Reporting basic demographic information such as age, gender, and additional health and mental health difficulties.
Utilizing standardized self-reported outcome measures to track the changes in severity of presenting symptoms	Reporting how COVID-19 was tested and managed, and how severe the symptom presentation was
Studying wider populations not only form one clinic or region	Reporting possible pre-existing associated factors such as local or systemic infections; vascular or autoimmune disorders; and stress, anxiety, and depression.
Undertaking audiometric assessments and comparing these with baseline audiograms or OAE results where available	Describing the tinnitus presentation such as its onset, frequency, descriptions, location, duration, and if it changes or resolves
Studies including longitudinal follow-up periods to identify the trajectory of the symptoms to indicate whether the tinnitus resolves or remain and if the severity changes	Investigating psychosocial factors that may contribute such as stress, anxiety, and depression
Providing management options to those presenting with audio-vestibular symptoms	Reporting tinnitus or auditory treatments offered and their effects

## Data Availability

Data is contained within the article and in the Appendix A.

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
