# Peer review of "The Impact of COVID-19 and the Pandemic on Tinnitus: A Systematic Review"

_jcm, 2021, doi:10.3390/jcm10132763_

Round 1
Reviewer 1 Report
Thanks for the opportunity to revise this interesting manuscript entitled “The Impact of COVID-19 and the Pandemic on Tinnitus: A Systematic Review” dealing with a topic of highly interest in the current literature. The language is fluent and reads well. The materials and methods were accurately planned according to the PRISMA guidelines. The manuscript can help in understanding the complex relationship between tinnitus and COVID-19.
Minor points to be discussed in the revised version:
- Why did you choose the “National Institute of Health Quality Assessment Tools”? This is a qualitative risk of bias tools while there are also quantitative ones as the STROBE score: 10.1097/MD.0000000000025917
- I would discuss more the pathogenesis of COVID-19 related audio-vestibular disorders in the discussion. Although it is just a speculation, there are some interesting studies in the literature dealing with this topic.
DOI: 10.1055/s-0040-1714370; DOI: 10.1016/j.amjoto.2020.102513
- Why did you not include the study of Mustafa Audiological profile of asymptomatic Covid-19 PCR-positive cases? I think it should be eligible.
- Another interesting point of discussion can be the difficulties faced by people affected by hearing loss during the COVID-19 pandemic.
DOI: 10.1016/j.amjoto.2020.102496 ; DOI: 10.1186/s12888-021-03118-0
Author Response
Author’s comment
Thank you for the useful suggestions. We appreciate the time taken to review this manuscript.
Reviewer comment
Why did you choose the “National Institute of Health Quality Assessment Tools”? This is a qualitative risk of bias tools while there are also quantitative ones as the STROBE score: 10.1097/MD.0000000000025917
Author’s comment
Thank you for this enquiry. There are different types of Risk Assessment Tools e.g. Ma et al. 2020). We sought one that was able to assess risk of bias in different study designs. From the shortlist, we selected the NIH to be consistent with the systematic reviews published on similar topics regarding COVID and the auditory system. We have now clarified this choice as follows in the manuscript on L133:
“Although other tools are available, using the same tool as used in similar systematic reviews [e.g. 10] allowed for consistency.”
Reviewer comment
I would discuss more the pathogenesis of COVID-19 related audio-vestibular disorders in the discussion. Although it is just a speculation, there are some interesting studies in the literature dealing with this topic.
DOI: 10.1055/s-0040-1714370; DOI: 10.1016/j.amjoto.2020.102513
Author comment
Thank you for this suggestion, we have now included this as follows:
L708 “Although speculative, numerous pathogenesis have been proposed regarding the possible association between hearing loss and the SARS-CoV-2 virus. Findings by Daikhes et al. [30], Freni et al. [43] and Desin and Pani [47] regarding reduced TEOAE amplitudes have been supported by Mustafa [59] who found that high frequency pure-tone thresholds and TEOAE amplitudes were significantly worse in 20 asymptomatic COVID-19 PCR-positive cases when compared with 20 normal non-infected participants. This indicates that the SARS-CoV-2 virus could affect cochlear outer hair cell functioning. Further suggested mechanisms suggest that the SARS-CoV-2 infection together with serotonin release and blood coagulation, may intertwine to activate platelets and drive SSNHL [60]. Excessive cytokine release and/ or ischemic damage from thrombosis, are furthermore suggested to increase oxidative damage resulting in permanent hearing damage [61]”.
Reviewer comment
Why did you not include the study of Mustafa Audiological profile of asymptomatic Covid-19 PCR-positive cases? I think it should be eligible.
Thank you for this suggestion. As this publication did not report tinnitus, it is not eligible as one of the included studies. We have however now included it in the discussion as follows:
“Findings by Daikhes et al. [30], Freni et al. [43] and Desin and Pani [47] regarding reduced TEOAE amplitudes have been supported by Mustafa [59] who found that high frequency pure-tone thresholds and TEOAE amplitudes were significantly worse in 20 asymptomatic COVID-19 PCR-positive cases when compared with 20 normal non-infected participants”.
Reviewer comment
Another interesting point of discussion can be the difficulties faced by people affected by hearing loss during the COVID-19 pandemic.
DOI: 10.1016/j.amjoto.2020.102496 ; DOI: 10.1186/s12888-021-03118-0
Author comment
Thank you for this suggestion. We have included this important point as follows:
“The wider pandemic effect, such as the impact of the use of non-transparent face masks hampering lip-reading and face coverings reducing the acoustic transmission, attenuating the sound, and preventing lip reading makes it difficult for those with auditory difficulties, especially those with greater difficulties e.g. cochlear implant users. A study of 59 patients with hearing loss attending hospital appointments in Italy, indicated that 37% reported moderate, 24% severe hearing difficulties [70]. These difficulties may contribute to the reports of increased anxiety during the pandemic for individuals with hearing loss, as demonstrated by a study focusing on 56 Iranian hard of hearing and deaf adolescence [71]. Support of those with hearing loss and other auditory symptoms is thus required”.
Reviewer 2 Report
See attached report.

Author Response
Summary: The manuscript contains a systematic review of studies having as (partial) topic the impact of COVID-19 on tinnitus. The manuscript is mostly well-readable, and the results are presented well. I have only minor comments mainly concerning style issues and suggestions requiring small changes. I am confident that only very little is needed for rendering the manuscript publishable.
Author’s comment
We appreciate the time you have spent on this manuscript, positive comments and helpful suggestions.
Minor points
- 2 l. 52: A CI ranging from 0.012 to -0.153 seems wrong.
Author’s comment
Thank you for pointing this out. This has been corrected as follows (L51): A further review by Jafari et al. [11] indicated a lower prevalence range (4.5%; CI: 1.2 to 15.3) from six studies.
Reviewer comment
- 3 l. 116: A reference for Rayyan is missing.
Author’s comment
This has now been added (L116) as: Johnson N.; Phillips. Rayyan for systematic reviews. J. Electron. Resour. Librariansh 2018, 30:1, 46-48.doi:10.1080/1941126X.2018.1444339
Reviewer comment
- 4 l. 149: Where is the value of 61% coming from? It seems a rather arbitrary choice.
Author’s comment
Thank you we have altered this as follows (L151): If I2 is high (larger value), indicating that effect sizes vary across the included studies, a random-effect model would be used to pool the data [25]. Borenstein, M., Hedges, L. and Rothstein, H., 2007. Meta-analysis: Fixed effect vs. random effects. Meta-analysis. com.
Reviewer comment
- 7 l. 268/9: You explain what betahistine is here. This should better be done when betahistine does first appear in the text (l. 227 p. 6).
Author’s comment
We have now clarified this as follows on first mention (L232):
“…betahistine, a dihydrochoride tablet often used to alleviate vertigo symptoms”.
It is also explained on Line 272 again:
“Cui et al. [27] reported tinnitus and dizziness for a 52-year-old male with positive coronavirus and a history of diabetes and Meniere’s disease, which was alleviated with betahistine, a commonly prescribed drug for balance disorders used to alleviate vertigo symptoms”.
Reviewer comment
- 7 l. 272: “with positive coronavirus” does not sound right. Is “test” missing? Same applies also at other occasions, e.g. P. 8 l. 293/4.
Author’s comment
This section has been reworded for clarity (L270):
“Vestibular difficulties associated with coronavirus were reported in only three patients (8%), all with positive results when tested for the coronavirus”.
Reviewer comment
- 10 Figure 2: The figure resolution seems a bit poor for figure of this size. Can the resolution be improved such that single dots are not visible anymore?
Author’s comment
We have used a higher quality image and hope this improves the resolution
Reviewer comment
- 11, hearing loss: the introduction and discussion of hearing loss could be structured better. First, hearing loss is commented on in l. 410. Then follows a paragraph starting with “Hearing loss:” in l. 419, followed by the “Section 3.5.6 Hearing loss”. I believe that readability would improve if all hearing loss related findings are reported together (e.g. in Sec. 3.5.6?).
Author’s comment
Thank you for this helpful suggestion. We have restructured this section as to include subheadings as seen on line 436.
Reviewer comment
- 13 l. 497/8: were both positive and negative PCR tests required in both studies? Or a positive test in one and a negative test in the other cited study?
Author’s comment
We have clarified this section as follows L526:
“Iqbal et al. [48] and Kamal et al. [53] only included participants who had PCR testing, to evaluate the presence or absence of SARS-CoV-2”.
Reviewer comment
Typos / phrasing
The language and grammar used throughout the manuscript is mostly of highly sufficient quality. However, there are some sentences where the clarity of the expression of ideas could be improved. Since I am not a native English speaker, please double-check if the items listed below require changes or not.
Author’s comment
Thank you for pointing these out, your English and proofreading is excellent, especially for a non-native speaker.
Reviewer comment
- 1 l. 13: missing “pandemic” after “COVID-19”?
Author’s comment
Corrected to: during the COVID-19 pandemic up to March 2021
Reviewer comment
- 6 l. 193: “,” after “[27]”?
Author’s comment
We have reworded this sentence for clarity to (L195):
“There were 35 cases in total with 9 case studies, 20 cases by Cui et al. [29] and 6 by Karimi-Galougahi et al. [30]”.
Reviewer comment
- 6 l. 236: “,” before “4”?
Author’s comment
Thank you, this comma has been added (L241): “…ten presented unilateral hearing loss, with 5 in the right ear [30,32,34], 4 in the left ear [30-31,33,38], and one patient presented unspecified hearing loss [39]”.
Reviewer comment
- 12 l. 483: “participants”?
Author’s comment
Thank you, this has been corrected to (L511): “In Viola et al. [41], 18.4% of the participants reported balance disorders”.
Reviewer comment
Table 2 / 3…: “Disor-ders” instead of “Dis-or-ders”?
Author’s comment
Thanks for pointing these out. The columns have been resized to avoid these additional hyphens.
Reviewer comment
- 11 l. 400: “location:”?
Author’s comment
This has been corrected to: Tinnitus location:
Reviewer comment
- 11 l. 409: “1 of 8”? Moreover, are the three patients with pre-existing hearing loss also part of the previously mentioned eight patients?
Author’s comment
This section has been clarified as follows (L415):
“In the study by Munro et al [40], there were eight individuals with tinnitus, of whom three also reported a pre-existing hearing loss. Of these, one participant reported that the tinnitus resolved over time”.
Reviewer comment
- 13 l. 493: “,” after “who”?
Author’s comment
This comma has been added as follows (L522):
Viola et al. [39] who, used an unspecified nasopharyngeal swab and Davis et al. [26] and…
Reviewer comment
- 15 l. 548: “score,”?
Author’s comment
Thank you, this comma has been added as follows (L577): “…improvements in the SAS, THI score, and tinnitus loudness test…”
Reviewer comment
- 16 l. 590: Remove the empty space before “=”
Author’s comment
This space has been removed
Reviewer comment
- 18 l. 645: what is meant by “-1-week post infection”? 1-week pre-infection?
Author’s comment
This has been corrected to 1 week
Reviewer comment
- 19 l. 717: is something like “attention” missing after “Raising a greater”?
Author’s comment
Thank you, this has been corrected to (L761):
“Raising greater awareness among healthcare providers is required…”
Reviewer comment
- 19 l. 721: “no as effective”?
Author’s comment
Thank you, this has been corrected to (L763):
“…that was normally helpful, was not as effective for those with bothersome tinnitus during the pandemic”.
Reviewer comment
- 19 l. 726: no “,” before “they”?
Author’s comment
This comma has been deleted.
Reviewer comment
- 19 l. 731: no “,” before “can”?
Author’s comment
This commas has been deleted
Reviewer comment
- 20 l. 744: “approaches”?
Author’s comment
This has been corrected to L789: “digital therapeutics approaches should be prioritized”
Reviewer comment
Page 3 of 3
- 20 l. 765: Since the possible relationship of ototoxicity and hearing loss / tinnitus is suggested here, there is a recent review by Tserga et al. (2019) on this subject (limited to cisplatin).
Literature
Tserga et al. (2019) https://www.nature.com/articles/s41598-019-40138-z
Author’s comment
Thank you, we have added this review as follows (L820):
“These may include the impact of receiving critical care including ototoxic medications [71], especially for those with a possible greater vulnerability to ototoxicity [72]”.